# Regional Spatial Structure and Land Use: Evidence from Bogotá and 17 Municipalities

**Andres Dominguez [1,\*], Hernán Enríquez Sierra [2] and Nicolás Cuervo Ballesteros [2]**

[1] Observatorio de Dinámicas Urbano Regionales, ODUR, Secretaría Distrital de Planeación, Bogotá 111711, Colombia

[2] Research Group on Public Policy and Business Economics, GIPE, School of Economics, Sergio Arboleda University, Bogotá 111711, Colombia; hernan.enriquez@usa.edu.co (H.E.S.); nicolas.cuervo@usa.edu.co (N.C.B.)

[\*] Correspondence: jdominguez@sdp.gov.co; Tel.: +57-3043942889

**Abstract:** The expansion of urban areas and the growth of the urban population are challenges faced by different territorial administrations across the world. In this context, the objective of this document is to analyze land occupation and the distribution of land uses in Bogotá and 17 municipalities. Therefore, a methodology is proposed in which an accessibility indicator models the spatial structure of the territory based on employment concentrations (the sum of the number of jobs weighted by the distance between each pair of municipalities). Then, the analysis of land use is carried out using a multinomial model, with the accessibility indicator as its principal explanatory variable. In such a way, it is possible to estimate the effects associated with the location decisions of economic agents in the territory. The results will enable policy makers to identify location and relocation patterns; we found evidence of a greater probability of commercial uses within urban areas and a relocation of industrial activity towards rural areas in some municipalities.

**Keywords:** spatial structure; accessibility; land uses; land use planning; urban sprawl



## 1. Introduction

Spatial structure is the pattern of distribution of economic activity in a territory. Therefore, the regional spatial structure can be analyzed from a morphological perspective, employing the concept of polycentric regions. From a conventional approach, this perspective seeks to establish how concentrations of economic activity are present in urban areas in different hierarchies and their effects on other variables of interest such as commuting, urban specialization, or land use. However, a broader sense of the term (which transcends the appearance of concentrations of economic activity within a city: intra-urban polycentrism), seeks to establish hierarchical relationships or interactions between these nuclei and nearby urban centers, encompassing the broadest analysis (interurban polycentrism) [1].

Polycentric regions may reflect spatial structures of economic activity concentrated in their main nucleus or with some degree of diversification in the location of activities. This is of particular interest in the way interactions between urban centers are generated and their level of diversification, which translates into the distribution of land uses in the study region. Polycentric regions also store information on their own evolution, from which questions relating to the size of the centers, their functional interconnection, and the regional scale of the polycentric phenomenon, among others, can be answered [1]. Although the original conditions of the region tend to be maintained in the case of relative population sizes or economic importance, the emergence and consolidation of sub-centers introduces structural changes, replacing the local functions of the dominant centers with new sectors (forming short-distance cities, metropolitan regions or mega-regions).

Polycentric regions develop a highly interconnected pattern from a central core. Moreover, in an advanced state of formation, centers are not easily physically distinguishable,

as they are administratively [2,3]. This modifies the original land uses as a response to greater economic externalities or regional specialization. Recently, the analysis of centers and sub-centers in appearance and functional relationships has been oriented to the interior of the most important nucleus of the city, from the theoretical development of the economies of agglomeration [4].

The evolution of monocentric regions towards polycentrism has been observed in developed countries [5,6]. However, in the case of regions in developing countries, where there has been urbanization without economic growth [7], it is interesting to explore why monocentric structures may persist in regions or where the transition to polycentrism has been faster but less functional than in developed countries. In Latin America, the transition towards polycentrism has been accelerated with the particularity that activities are concentrated within the main urban centers, giving rise to sub-centers of employment within the larger cities and a smaller formation of nodes in the metropolitan periphery [8]. Regions relevant to the Latin American context, such as Santiago de Chile [9] or the city of Buenos Aires [10] and their respective nodes, present morphological patterns that have evolved from a fragmented or functionally dispersed region to a polycentric region at both the intra- and interurban scale.

This document seeks to advance the research about regional polycentric structures in less developed countries from the perspective of land use. With this objective in mind, this study examines the relationship between the spatial structure and land uses in different sectors of the Bogotá Metropolitan Area. The interest in this case is to identify the underlying pattern, considering rapid urbanization, added to the effects of overflow from the main city. The Bogotá Metropolitan Area is defined as Bogotá and 17 surrounding municipalities (Bojacá, Cajicá, Chía, Cota, Facatativá, Funza, Gachancipá, La Calera, Madrid, Mosquera, Tabio, Tenjo, Sibaté, Soacha, Sopó, Tocancipá and Zipaquirá) which are not a legal or administrative unit but are considered as a functional integrated area by the National Statistics Office and used in the literature [11]. This functional integration is also recognized in the spatial coverage of statistical sources such as the Bogotá Multipurpose Survey and the Bogotá-Regional Economic Enterprise Survey (EEEBR 2017), the last of which is used as a source in this research.

A contemporary quantitative approach, employed in this study, is oriented towards the distribution of residential and economic activities in space rather than an absolute comparison of the centers and sub-centers of economic activity. This approach focuses on spatial techniques and scenario analysis to determine the relative importance of the centers [12,13]. It also allows for the establishment, in morphological terms, of an evaluation of the conditions of a region as a whole, and the requirements that are needed to guide the trajectory of a polycentric region through the identified determinants which explain land use. The empirical analysis is carried out at the scale of city blocks in urban areas, and land plots or lots in rural areas, providing the results with an important degree of detail.

In specific terms, this document seeks to explain the distribution of land uses in Bogotá and its 17 surrounding municipalities through various determinants, giving priority to the level of accessibility as an expression of the regional spatial structure. The analysis is broken down by various uses, which seek to verify whether economic activities associated with commerce and services have greater incentives to locate near sites with higher levels of accessibility. The analysis also considers whether they do this by exercising competition for land use rather than, for instance, activities associated with industry or residential use that would be located in areas with medium or low accessibility. This paper thus takes up a specific characteristic of descriptive models of land use change, highlighted by [14]. It starts from the point of view of a landowner who obtains rent from the land by allowing another agent to occupy it, with the aim of maximizing the expected returns or utility derived from the land.

The estimation of land use is developed with the specification of multinomial models, whose main explanatory variable is the level of accessibility. This accessibility variable is calculated as the sum of the number of jobs in each spatial unit, weighted by the distance

between each pair of the spatial units. The main results include a greater probability of commercial uses within urban areas and a shift in industrial activity towards rural areas in some municipalities other than Bogotá.

The perspective adopted relates to research on the effect of transport infrastructure on land value and use. As observed by [15], transport infrastructure investments generate changes in land values, which, in turn, are reflected in land use change. In this context, [16] investigated the impact that subway stations have had on the amenities of nearby neighborhoods, because the construction of a station increases the number and diversity of restaurants in nearby areas. The authors of [17] showed how train stations not only increase property values, but also attract commercial economic activity, which means changes in land use. Land use analysis also has contributions from another, broader scale. For instance, [18] analyzed uses associated with settlements (crop cultivation, roads, and pastures, among others) in the period 1993–2008 for a province in Beijing. The authors identify the determinants and changes in land use through the estimates from a multinomial logistic model.

After this introduction, this paper continues with a review of the literature on this research topic. The following section presents the theoretical model; the basis for the empirical analysis carried out in this document. The section contains the empirical analysis of descriptive statistics, the calculation of the accessibility variable, and the explanation of control variables. The fourth section presents the estimation model, and next we discuss of the results. The last section presents the main conclusions.

## 2. Literature Review

In market economies, the combined effect of market forces and administrative decisions determine spatial structures [19]. Taking advantage of the primacy of location and market growth, based on local environmental and physical conditions, it gives rise to polycentric regional structures in both morphological and functional terms. In the literature, the term 'morphological' is related to the relative balance of cities across economic or demographic variables. In the end, a polycentric region is an integrated system that is more than the sum of its parts [20], concerning the human activities which take place in kinds of agglomeration patterns through space.

The concentration of economic activities in specific zones of urban areas creates an unequal spatial distribution of land uses, with consequences that manifest themselves outside their boundaries. The willingness to pay for land associated with different activities is the mechanism by which competition for proximity to the centralities is resolved. In this sense, as [19] argues, in urban environments markets act through land prices. Locations where the price of land is high (e.g., due to high demand or land scarcity) lead to land being used intensively (for instance, tower blocks with a greater number of floors). However, in locations with lower land prices, more extensive use is made of land; therefore, the density of construction will be lower. This generates tensions in land use due to competition between economic activities. This competition occurs for goods where supply and demand are different when compared to other goods and services in an economy. In specific terms, each lot of land has a fixed location that is associated with unique characteristics in terms of soil quality, slope, altitude, accessibility, etc. [21–24].

Although there is a vast amount of literature related to urban spatial structure using methodologies for identification, it is not the same as from a metropolitan or regional perspective. An analysis based on the functionality of a region focuses on the integration of the main city with secondary cities, and of the secondary cities with each other [6,25], although the quantitative identification or description of such an intercity spatial structure has a limited volume of studies.

From this point of view, the analysis of densities (of population, built space and employment) allows for the analysis of urbanization patterns (sprawl and mega-urbanization) [26] and the characteristics of regional economic development (development hubs, nursery cities and megacities) [27–29]. The basis for interest in regional analysis is rooted in the

fact that positive externalities and institutional arrangements make each urban component of the region more competitive, more socially cohesive and more environmentally sustainable [30,31].

The literature has identified and classified the causes of agglomeration using concepts such as technological spillover, labor pooling and intermediate input linkages [32,33], or as mentioned by [34], sharing, matching and learning effects. Therefore, significant levels of concentration of economic activity generate externalities which are leveraged by companies and reflected in increases in productivity, and in workers' wages [21,35]. Similarly, there is evidence that confirms the positive relationship between productivity levels and levels of concentration of economic activity [36–41]. As the literature highlights, this relationship exists both for different economic sectors and for different levels of spatial aggregation.

In this context, the literature has advanced in two directions: the identification of regional centers and sub-centers and the effects of their existence on activities and land uses in neighboring areas. The identification of polycentrism is based on an interest in showing whether there is a relative balance between regional centers in terms of their importance [30]. The empirical approaches are based on the measurement of two aspects: the importance of each center and how to evaluate the importance in each center [42]. Traditional measurement methods rely on the comparison between the different centers using absolute indicators such as population size or participation in the regional product, or relative measures such as size-range [43,44]. Taking into account studies such as [45], the search for a robust measure of spatial structure that includes both economic and physical characteristics of system of cities is represented by statistical models rather than index or other indicator descriptions.

The identification of areas with high densities and concentrations of employment (employment over area) enables us to analyze the patterns of centrality or clustering, made up of a center of major importance and less important concentrations of economic activity that, in some cases, complement the main center, and in others substitute its functions to a lower hierarchy. An analysis of these characteristics oriented towards planning seeks a balance between the location of jobs and residences. This could be encouraged through the provision of public services in specific areas, or through actions that incentivize either the densification of existing centers or set parameters so that their expansion does not lead to changes of use that are considered undesirable.

In particular terms, the empirical analysis of the urban structure of the city of Bogotá has been addressed in some studies, as well as the economic effects on cities in Colombia from a regional perspective. For instance, [46] identified the center and sub-centers of employment in Bogotá. Additionally, using the methodology proposed by [47,48], ref. [49] identified the employment center and sub-centers where economic activity is concentrated in Bogota—employment records for the year 2005 were used, provided by the city's District Planning Office. The authors used descriptive statistics to identify the center and sub-centers of economic activity. Specifically, they identified those sectors where the density of employment is greater than or equal to the average density of the city. Additionally, the level of employment was greater than or equal to 1% of the total employment in the city. The CBD, or Central Business District, is geographically identified as being in the traditional center of Bogotá. There is also evidence of the existence of the following sub-centers: San José, Álamos (to the west, near El Dorado Airport), and a further three to the north of the city: Avenida Chile, Calle 100 and Calle 127.

Methods derived from the literature concerned with the identification and classification of regional centers and subcenters have limited pertinence to address these issues; the theory and methods presented in the next section tackle the influence of the existence of regional centers and subcenters considering the rationale of economic agents and by constructing an accessibility indicator available both for zones characterized as regional centers and subcenters as for those that are not, where the land use change is also of interest.

## 3. Theory and Methods

### 3.1. Theoretical Framework

The objective of this section is to consider the principal theoretical aspects under which it is possible to analyze the functional pattern of a region composed of multiple cities. These are centers that seek to reap the benefits of agglomeration and lower costs of market access, either by the traditional route of transport or by access costs. To this end, the key theoretical elements for analyzing polycentric effects on two scales are presented. In the first, the forces and nature of agglomeration are examined, which leads to the generation of a center or centers of activity from which the main aspects of the urban structure can be configured.

Among the most important theoretical contributions to the analysis of the spatial structures of cities are those made by [50–52]. They discussed how the locations of businesses and residential units were established, starting from the gravitational influence that a center of economic activity, such as the Central Business District (CBD), can generate. These kinds of models start from the deterministic location of the CBD and the distance to any point in the urban area as a fundamental variable to explain agents' willingness to pay for the use of the land [53].

From these theoretical developments, [54] developed a linear model in which a deterministic CBD was not assumed. The authors incorporated the positive impacts generated by the economies of agglomeration as the main generator of concentrations of economic activity. Positive externalities enable companies to be more productive, and this incentive makes the existence of a CBD possible. Initially, it is low hierarchy, but as the concentration of economic activity becomes more important, the spatial structure of urban environments is progressively defined. Therefore, the CBD has a direct influence on land income, because competition for land near the CBD creates conditions of scarcity.

A modification to the [54] model was made by [55]. In this case, the authors developed a circular model in which both companies and households compete for land use. The economies of agglomeration generate incentives for companies to concentrate in certain parts of the geographic space, and competition for the use of land close to these locations displaces residential activity. For households, the main variable is related to the travel costs workers must incur to reach their place of work. Therefore, companies have incentives to locate close to other companies, whereas households seek residential locations where travel costs are minimized. In this way, land rents and workers' wages influence the location decisions made by companies and households.

In formal terms, companies that produce goods and services require land and workers. Given that the productivity of companies has a positive relationship with the levels of economic activity that are concentrated around them, the production per unit of land in location $s$, $x(s)$, is expressed as follows:

$$x(s) = A\, z_s{}^\gamma n^\alpha \tag{1}$$

In this specification, $A$ is a productivity constant, and $z_s$ represents the effects of spatial agglomeration on the production levels of location $s$. This agglomeration effect is a variable that is calculated by adding the total employment in the locations $s$ and weighted by the distance to each of these locations. It is also necessary to highlight that this variable is interpreted in terms of accessibility to employment, measured as concentrations of economic activity. The number of workers in each company is represented by $n$, and the parameter $\alpha$ is less than one.

The profit of the company per unit of land at location $s$, $q(s)$ is formally represented as follows:

$$q(s) = A\, z_s{}^\gamma n^\alpha - w(s)n \tag{2}$$

where $w(s)$ represents the wage that companies can pay in location $s$. In this way, each company chooses a degree of employment $n$ that enables it to maximize the profit function. The first-order conditions allow us to obtain the optimal values of $n = f(w, z)$ and

$q = f(w, z)$, which results in the maximization process. In other words, given wages $w$ and the levels of agglomeration, $z$, the willingness to pay for each unit of land is obtained.

In the case of the residential location of households and workers, a utility function is proposed that depends on a consumer good ($c$) and the amount of residential land ($l$):

$$U(c, l) = c^\beta l^{1-\beta} \tag{3}$$

The transport costs that the worker must incur to travel from their place of residence, $r$, to their place of work, $s$, is formally described as:

$$e^{-\kappa|r-s|} \tag{4}$$

In this way, the salary that the worker receives is multiplied by the cost of transportation:

$$w(r) = e^{-\kappa|r-s|} w(s) \tag{5}$$

where $\kappa$ represents the spatial friction, which depends on the transport system. This is how the model conditions workers to make residential location decisions.

Therefore, land is assigned to localize economic activity and residential activity. The model allows for the acquisition of income gradients. This means that in areas with the highest density of economic activity, land prices are higher and higher wages can be paid. On the other hand, in residential areas closer to amenities or positive externalities where salaries remain high, residential land prices will also be high. The spatial distribution of land uses (economic and residential) will depend on which of the curves of willingness to pay dominates.

Therefore, the theoretical model shows that in cases where the gradient of economic activity dominates, zones of exclusive economic use will be generated. On the other hand, when the gradient associated with residential activity dominates, exclusive areas of residential use will be generated. Finally, when the gradients are similar, mixed-use zones will be generated. The urban structure in this model does not necessarily predict that residential areas are concentrated on the edges of the city and that business areas are concentrated in the center. As in monocentric regional structures, it allows for the possibility that there are several centers of activity and areas of mixed economic and residential use. In fact, several studies have shown evidence of how monocentric structures have been transformed into polycentric or dispersed structures [3,48,56].

*3.2. Characteristics Specific to Bogotá and 17 Municipalities*

In empirical terms, the regional analysis covered by this document includes Bogotá, the most important city in Colombia in economic and population terms (generating approximately 25% of GDP according to 2017 figures, with 16% of Colombia's total population according to the 2018 population census), and 17 neighboring municipalities (with which Bogotá has been shown to have a commuting relationship for work or study reasons): Bojacá, Cajicá, Chía, Cota, Facatativá, Funza, Gachancipá, La Calera, Madrid, Mosquera, Soacha, Sibaté, Sopó, Tabio, Tenjo, Tocancipá and Zipaquirá.

Bogotá's population grew from 4,947,890 in the 1993 census to 6,707,338 in 2005, a growth of 38%. The 2018 population census data recorded a population close to 7,421,566, which represents a growth of 10% with respect to 2005. The urban area of Bogotá grew from 28,153 hectares in 1991 to 38,683 in 2005, a growth of 37%. By 2018, the urban area had reached 41,696 hectares, a growth of 7.8% between 2005 and 2018. In conclusion, according to information from the National Population and Housing Census (CNPV, 2018), Bogotá has a population of 7,421,566, and the 17 municipalities in Cundinamarca have a combined population of 1,712,578, making a total of 9,125,144 inhabitants for the study area.

Urban growth and the regional connectivity network have been limited to connections with the main city, which has generated two phenomena of interest for this analysis. The first is that there are municipalities that have now formed conurbations with Bogotá,

such as Soacha, as well as conurbations that have developed between peripheral municipalities, such as Funza–Mosquera. The second is where a road network connects the main node, in this case Bogotá, which, in some cases, limits links between municipalities. Although this does not mean having to travel through the city to achieve regional connectivity, it does determine mobility patterns in the region.

It is important to note that the study region is not jointly planned or organized as a whole. That is to say, that most of the phenomena observed in the territory are largely due to dynamics generated by the economic influence of Bogotá and the local interests of attracting economic and residential activities. This results in a heterogeneous and fragmented urban growth process, prone to generating phenomena such as socio-spatial segregation and functional specialization of the nodes in certain activities such as real estate or industry. The nature of the interdependence with the capital as the main market, however, has led to a shift from a planned orientation to a disorderly competition for strategic locations, such as main roads.

The map in Figure 1 displays the urban area of the region, which shows how functionality is expressed in the occupation of land with accessibility criteria of the main roads and to Bogotá. It should also be noted how particular interests are expressed in urbanization processes, where suburban land is prioritized in the west and north of the region more than in the south or center of Bogotá.

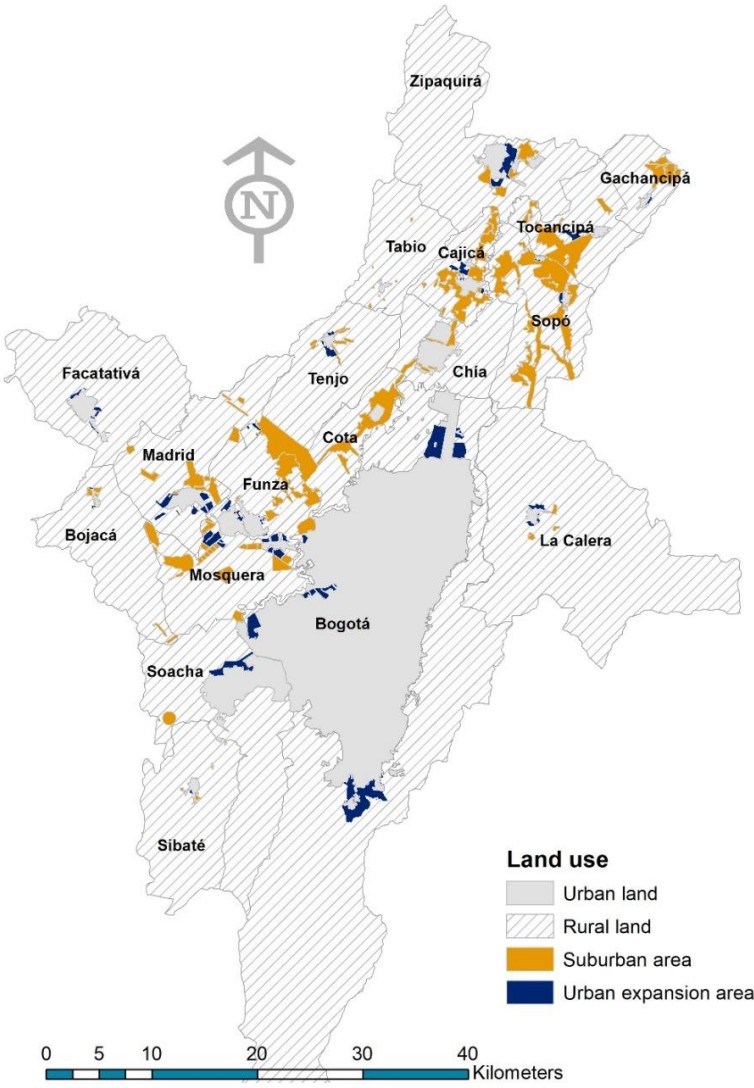

**Figure 1.** Current land classification. Based on information from [57].

The distribution of economic activity can be seen by the destinations of commuters which are recorded in mobility surveys, as carried out by [58]. Figure A1, in the Appendix A, shows how when mapping the destination of work journeys or commutes per hectare, there are a significant number of areas in Bogotá with a density of work destinations, as well as in the municipalities of Soacha, Cota and Tocancipá. However, commuter destinations alone are not a sufficient indicator of the regional economic spatial structure; one reason being simply that mobility surveys do not have this objective.

In terms of land use, it is possible to calculate the predominant use in each block or plot of land in the study area, using the percentage of square meters built per use (recorded in the cadastral records in 2017). Table 1 shows the distribution of land uses in the region for several categories, the main five uses are: Residential, Commercial, Industrial, Public Use and Agricultural.

**Table 1.** Number of spatial units by predominant land use in Bogotá and 17 municipalities.

| Use | Zone | | Total |
| --- | --- | --- | --- |
| | **0—Rural** | **1—Urban** | |
| No Use | 36,453 | 647 | 37,100 |
| Residential | 18,162 | 41,605 | 59,767 |
| Commercial | 873 | 3528 | 4401 |
| Industrial | 554 | 162 | 716 |
| Public Use | 942 | 1565 | 2507 |
| Agricultural | 21,606 | 4 | 21,610 |
| Other Uses | 160 | 3763 | 3923 |
| Urbanizable | 37 | 2 | 39 |
| Urbanized | 18 | 1 | 19 |
| Not Urbanizable | 8 | 2 | 10 |
| TOTAL | 78,813 | 51,279 | 130,092 |

Source: Based on information from the IGAC and the Cadastral system of Bogotá (2017).

Figure 2 shows the distribution of land uses in the study area. The category 'no use' indicates that in the spatial unit there is no construction of any type. Gray identifies the spatial units with square meters constructed for residential use; blue represents commercial use; yellow is industrial; and orange public land. Table 1, alongside the cartographic information, shows the number of spatial units per use in the study region, controlled for the zone variable which takes a value of 1 in urban areas and 0 in rural areas.

*3.3. Job Accessibility*

In this section, we analyze the way we identify the accessibility to jobs in the study's research area that includes Bogotá and 17 neighboring municipalities. The information used is drawn from the 2017 Bogotá-Regional Economic Enterprise Survey (EEEBR 2017), carried out by the Bogotá's Planning Office. Cadastral information was taken from the Agustín Codazzi Geographical Institute (IGAC) for the 17 municipalities.

The accessibility indicator used in this paper was constructed following the [55] model and the contribution of authors such as [59,60]. The indicator $Z_s$ represents the level of economic agglomeration, or accessibility to employment, associated with the spatial unit $s$ taken from the employment located in this same area and also from all the other locations that are part of the study area. In formal terms, it is as follows:

$$Z_s = \sum_{k=1}^{S} e^{-\delta m_s^k}\, \widetilde{E}_s \; ; \; \delta \geq 0 \qquad (6)$$

where $\widetilde{E}_s$ represents the amount of employment in a location $s$, $m_s^k$ is the distance between different spatial units $k$ and $s$, and the parameter $\delta$ defines the speed at which the effect of spatial agglomeration declines with distance (it is a spatial friction parameter).

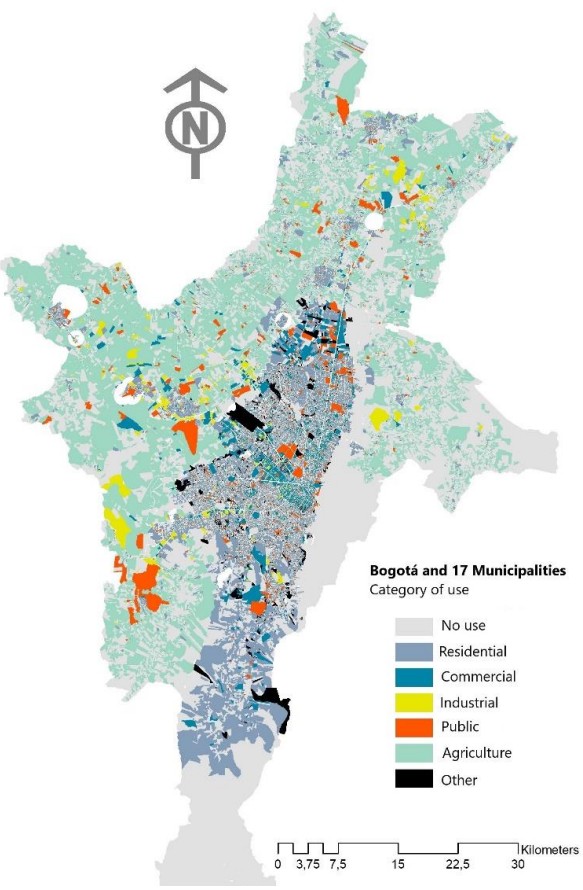

**Figure 2.** Predominant land use in Bogotá and 17 municipalities. Based on information from the IGAC and the Bogota District Cadastral.

The EEEBR (2017) provides the data to calculate the accessibility to employment indicator, an input for the analysis of the determinants of land use in different sectors of the metropolitan area. The land use analysis uses cadastral information, the accessibility indicator as well as complementary information (e.g., distance from main roads, reserve areas) to study the determinants of use, the differences for locations with different characteristics, and to explore the possible changes of use in the study region. This section describes the main methodological elements used to conduct the research, and the following section gives the results of the land use analysis.

It is important to consider that the spatial effects are incorporated in the calculation of this variable, in addition to employment levels. This is because the main input is the matrix of origin destination, which includes the distances between the spatial units of analysis in its rows and columns. Therefore, the friction parameter $\delta > 0$ and the distances between the spatial units are key because they calculate the transportation costs in the study region. When these measures are modified, the sum totals change, allowing this fundamental variable in the regional structure to change. It is possible to evaluate the extent of the spatial effect or accessibility levels between spatial units with different values for the friction parameter. In the econometric exercise, the appropriate friction parameter will depend on the fit statistics calculated for the regression models that need to be estimated. It should be noted that a friction parameter equal to 0.5 or 1 allows the extent of agglomeration to be greater in study area. When the parameter is greater (for instance equal to 5), the effect is

not as far-reaching. Figure 3 shows the accessibility calculation with a friction parameter $\delta$ equal to 3.

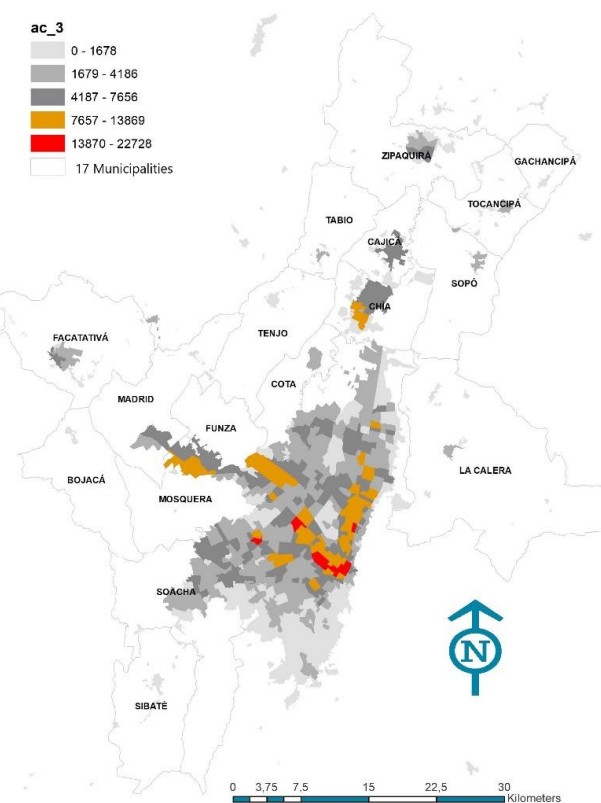

**Figure 3.** Accessibility indicator with friction parameter equal to 3. Based on information from the EEEBR (2017).

The pattern of regional accessibility shows a concentration of employment, and therefore economic activity in the regional area. The highest values for the indicator are found in Bogotá as the main city and include most of the agglomerations previously found in the literature. Specifically identified are the business and industrial centers, commercial zones and the airport.

Additionally, this indicator allows for the identification of two patterns: first, it is possible to identify in 2 of the 17 municipalities an important value for the indicator (which, compared to that observed in the interior of Bogotá) that increases the importance of these urban centers: Chía and Mosquera (Chía, where activities associated with commercial and public land uses are located; Mosquera, with a predominance of industrial land use). The two municipalities each display differentiated characteristics, in addition to the predominance of the land uses mentioned.

The pattern not observed according to the indicator's lowest values are for the Soacha and Tocancipá municipalities, where the region's industry has concentrated. However, they do not exhibit accessibility levels that could lead them to be considered as urban centers as important as those mentioned above, in terms of economic activity.

The accessibility indicator used in this study differs from traditional techniques that apply statistical criteria to identify those spatial units with the highest amount and density of employment. The traditional method was not considered because it does not take into account the underlying spatial effect. In other words, in most cases, both the employment center and employment sub-centers are surrounded by spatial units with significant concentrations of economic activity.

To compare our accessibility indicator with the identification of centers and sub-centers, we used the methodology proposed by [49] for the Bogotá metropolitan area with

data from the EEEBR (2017). This registered 1,541,088 jobs and an average employment density at the census sector level of 33.7 jobs per hectare. Figure 4 shows the census sectors that have above average densities (1) and those that (in addition to exceeding overall average density), have an employment percentage greater than 0.8% (2), which are cataloged by this methodology as employment centers. There is only one census sector that exceeds 1%, and that is Corabastos (1.04%). The other census sectors that meet these two conditions are on the road corridor that connects the city's traditional center (CBD) with El Dorado Airport.

In summary, the spatial structure of the region with Bogotá as the main urban center contains a group of zones of concentrated economic activity with a greater predominance of employment, although this also includes nearby municipalities. However, it is possible to distinguish how this distribution of concentrated economic activity does not cover the entire city, nor all the municipalities. In some of the following, concentrations of economic activity are minimal or practically non-existent, as in the cases of Gachancipá, Tabio, Tenjo, Bojacá and Sibaté. These patterns are better identified by the accessibility to jobs indicator than the usual job density measures.

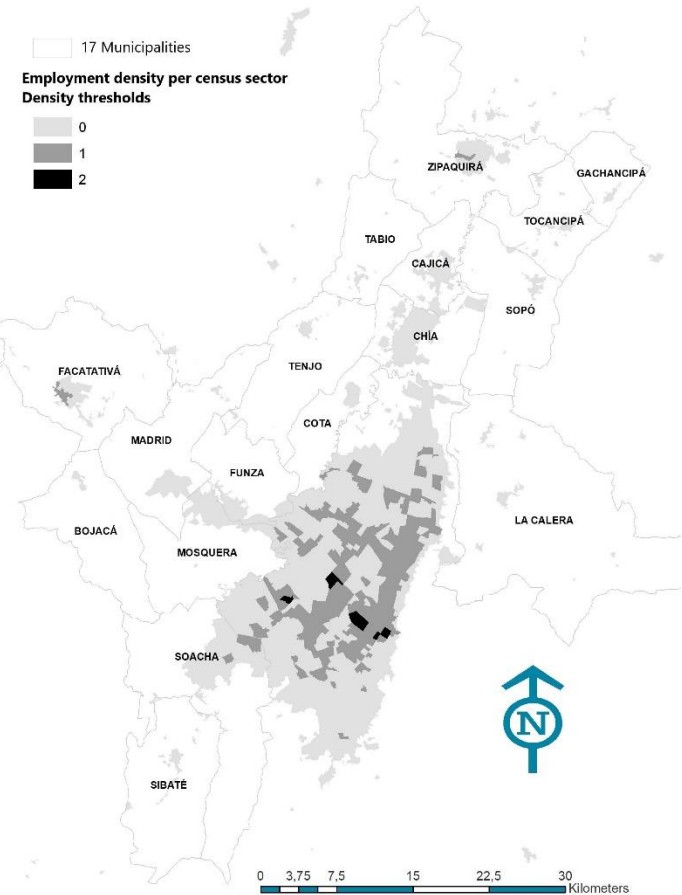

**Figure 4.** Employment densities in Bogotá and 17 municipalities, 2017, using methodology proposed by [44]. Based on information from the EEEBR (2017).

### 3.4. Methodology

The values obtained by the accessibility to jobs indicator contribute to the analysis of the determining factors of land use. The application of discrete choice models for land use is explained in the specialist literature as a tool to explain choices between mutually exclusive alternatives [61]. These econometric models with dependent variables of more than two options are generally estimated by maximum likelihood. In this paper, the dependent variable represents the alternative land uses. From an economic perspective, it is proposed

that one of the categories of land use is related to a higher level of utility or benefits, compared to the rest of the alternatives.

To formalize the model, this methodological approximation explains the probability that a land use will materialize taking all other land use possibilities as a reference [45]. The model specification takes the following form:

$$P_{iu} = \Pr[y_i = u] = \frac{e^{(\beta_1 * x_{i1} + \beta_2 * x_{i2} + \ldots + \beta_k * x_{ik})}}{\sum_U e^{(\beta_1 * x_{i1} + \beta_2 * x_{i2} + \ldots + \beta_k * x_{ik})}} u = 1, \ldots, U \tag{7}$$

where the dependent variable $P_{iU}$ represents the probability that spatial unit $i$ has a land use $u$; $e$ represents the base of the natural logarithm; $x_{ik}$ represents the different explanatory variables that are used to model the suitability of the spatial unit $i$ to accommodate the use $u$; and finally, the $\beta$ values are the parameters that fit the model.

## 4. Results

### 4.1. Land Use and Regional Spatial Structure

This section presents the results of a model that estimates land uses in the spatial structure identified by the accessibility indicator. The results make it possible to determine the regional functional relationship between Bogotá and its sub-centers and the activity centers found in each of the municipalities. At the end of the section, the spatial trend of the data is discussed in light of the observed spatial pattern.

The model estimated with the specification of Equation (7) for the study area allows the calculation of the probability that a land use is located in a specific area of the regional spatial structure (Figure 2). This takes into account spatial characteristics that we include as explanation variables in the econometric model, such as accessibility conditions, distance from main roads and the following additional control variables: binary variables for expansion zones and suburban zones of Figure 1 (which allow for the consideration of potential growth in the urban footprint of each municipality); geographical restrictions, based on information from the IGAC regarding highland "paramo" areas, forest protection areas and freshwater bodies and the main restrictions for growth of urban areas in the municipalities; the urban/rural variable that takes the value of 1 in urban areas and 0 in rural areas; and the coordinates (X, Y) that are included as a control for unobserved spatial effects.

Table 2 shows the results of the estimation of two models with the multinomial logit specification, as shown in Equation (7), to estimate the probability of land use according to a set of explanatory variables. The categories of the dependent variable coincide with the most relevant of the first column in Table 1, and are residential, commercial, industrial, public and agricultural. In the interpretation of the estimates, the coefficients associated with each variable indicate a greater or lesser probability of use corresponding to the indicated category with respect to residential use (base category). The choice of the base category does not modify the probability calculation (shown below) from which the spatial analysis of the results is made.

It should be noted that models of this type must meet a test of irrelevant alternatives. In other words, excluding one of the categories in the dependent variable should not generate statistically significant changes in the estimated coefficients. Therefore, a Small–Hsiao test was conducted (in which if the null hypothesis is rejected, it means that the elimination of one category of the dependent variable is not independent of the others). The result of estimates where a category is excluded, and the coefficients of the unrestricted model are compared, is that the differences between the coefficients of the two models are not significant. In that way, it is not possible to reject the null hypothesis and it is assumed that the alternatives are independent (see test results in Figure A2 in the Appendix A).

As can be seen (Table 2), the estimates for two spatial accessibility indicators show the robustness of the results in relation to two friction parameters. The first, when the spatial friction parameter is equal to 0.5, and the second, when this parameter is equal to 2 (see maps in Figure A3 in the Appendix A). The results show that the coefficient associated

with the agglomeration indicator is positive in both cases and its value decreases as the friction parameter increases, which is consistent with the intuition and theory presented in this paper. Furthermore, as shown by the fit statistics for each of the models in the lower part of Table 2, the differences are not overwhelming.

**Table 2.** Estimation results of the multinomial model of land use.

| | log(access_0_5) | | | | log(access_2) | | | |
|---|---|---|---|---|---|---|---|---|
| | **Com.** | **Ind.** | **Dot.** | **Agr.** | **Com.** | **Ind.** | **Dot.** | **Agr.** |
| Log(acc) | 1.06 *** | −0.03 | −0.02 | −0.38 *** | 0.95 *** | −0.19 *** | −0.03 | −0.22 *** |
| Log(dvias) | −0.46 *** | 0.16 *** | −0.04 * | 0.34 *** | −0.50 *** | 0.18 *** | −0.04 * | 0.34 *** |
| Urban/Rural | −2.08 *** | −1.68 *** | −0.29 *** | −8.38 *** | −0.54 *** | −1.48 *** | −0.31 *** | −9.13 *** |
| Expansion | −0.18 | −1.08 *** | 0.04 | −1.11 *** | 0.83 *** | −1.06 *** | 0.02 | −1.34 *** |
| Suburban | −0.01 | 2.14 *** | −0.45 *** | −0.38 *** | 0.10 | 2.18 *** | −0.45 *** | −0.51 *** |
| Forest Reserve | 0.32 *** | −1.84 *** | 0.30 *** | 0.18 *** | 0.71 *** | −1.89 *** | 0.30 *** | 0.22 *** |
| Water Reserve | −0.32 *** | −0.11 | −0.21 * | 0.13 | −0.19 * | −0.08 | −0.21 * | −0.14 |
| Coordinate X | ✓ | ✓ | ✓ | ✓ | ✓ | ✓ | ✓ | ✓ |
| Coordinate Y | ✓ | ✓ | ✓ | ✓ | ✓ | ✓ | ✓ | ✓ |
| Constant | −11.15 *** | 0.20 | −2.63 *** | 1.84 *** | −9.02 *** | 0.81 * | −2.63 *** | 0.42 *** |
| Number of obs. | 88,987 | | | | 88,987 | | | |
| LR chi2(44) | 51,325 | | | | 50,455 | | | |
| Pseudo R2 | 0 | | | | 0 | | | |
| Log likelihood | −54,343 | | | | −54,777 | | | |

*** significant to 1% * significant to 10%; ✓ econometric models include the variables as control variables.

The results show evidence about increases in the accessibility variable; for instance, those with a friction parameter equal to 2 (Log (acc)) are related to changes in land use from residential to uses associated with trade and public use. The sign associated with the coefficient of this variable is negative with respect to industrial and agricultural use.

The coefficient associated with the distance from the road network (Log (dvias)) also generated the expected sign. In other words, increases in the distance from district, regional or national roads are related to a decrease in the probability that the land will be used for commercial or public activities. The sign is positive with respect to industrial and agricultural use (which makes sense according to the dynamics generated by the willingness to pay for land use) within the competition to which economic agents are subject, based on which trade has a greater willingness to pay compared to industry and agriculture.

The spatial expression of the probabilities of use obtained through the model can be found throughout the region. There is a high probability of residential use in the urban areas, and the municipal configuration of uses differs between municipalities in commercial and industrial use.

Table 3 summarizes the average probability that each of the spatial units in the region (blocks in urban areas, lots in suburban and rural areas) take a specific use from those studied in this paper. As expected, the highest probability is for residential use. What is of interest is how the other uses analyzed are distributed in relative terms.

Agricultural use is ranked second in average probability, which can be explained by the size of the region and the structure of accessibility for the municipalities. The latter is because, given the other controls used, the extent to which a piece of land has less accessibility, the more likely it is to remain in rural use. Commercial and public uses have a probability that (although not high when compared to residential and agricultural uses), is higher than that of industrial use. This highlights one of the conditions of the region,

which is characterized by a relatively widespread presence of commercial and public areas, whereas industrial activities are concentrated in just a few areas.

**Table 3.** Average probabilities of land use.

| Use | Probability | Std. Dev. |
|---|---|---|
| Residential | 60% | 26% |
| Commercial | 4% | 6% |
| Industrial | 1% | 2% |
| Public | 3% | 1% |
| Agricultural | 32% | 29% |

Calculations for 130,065 spatial units.

To complete the description of these results, maps in Figure 5 show the probabilities estimated for each of the land use categories considered in the multinomial regression model. The results show that the probability of residential use is relatively high in urban areas in municipalities, as would be expected. Red identifies the spatial units where the probability exceeds 80%. Orange identifies the spatial units where the probability is between 60% and 80%.

Bogotá concentrates the highest probabilities in the predominant areas of commercial use. What is striking in this case is the widespread appearance of this probable commercial use in the Chía, Mosquera, Soacha and Zipaquirá municipalities. The first two were identified in the accessibility pattern by employment, whereas the others were not identified as significant centers of economic activity.

In the case of industrial use, a pattern of probabilities is presented that is dominated by the municipalities of Mosquera, Soacha, Funza and Cota. In addition, there are possible patterns of occurrence in municipalities farther away from the main nucleus, such as Sibaté, Bojacá, Facatativá, Cajicá, Sopó, Tocancipá and Gachancipá.

Finally, for public and agricultural uses, the expected behavior was observed. For the first use, a greater probability was observed in urban areas, whereas for the second, it was observed for areas outside urban areas and far from road corridors.

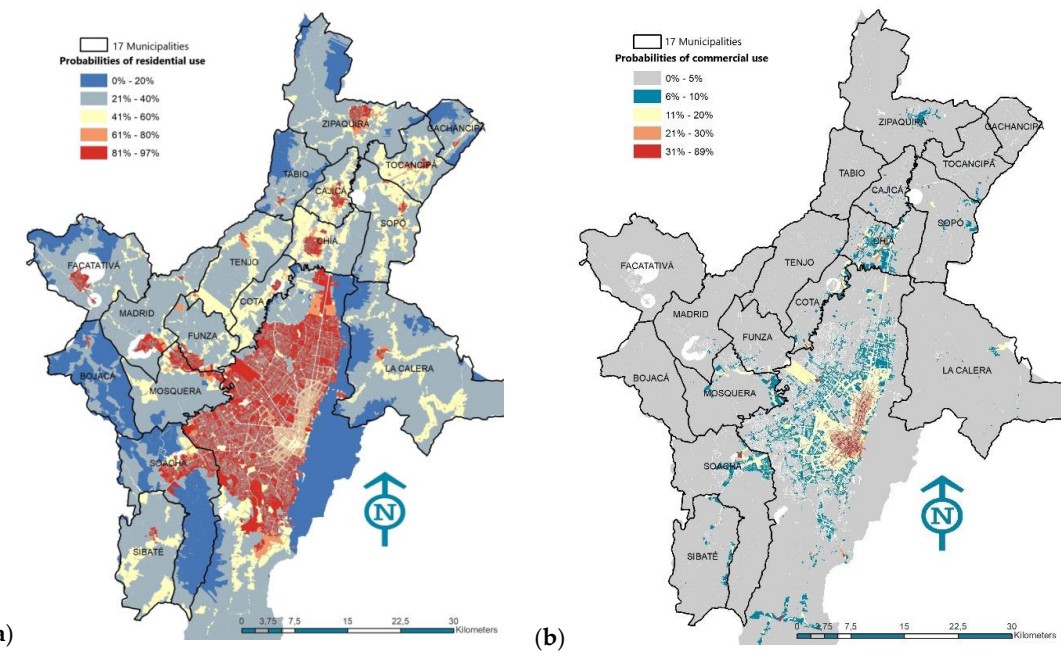

**Figure 5.** *Cont.*

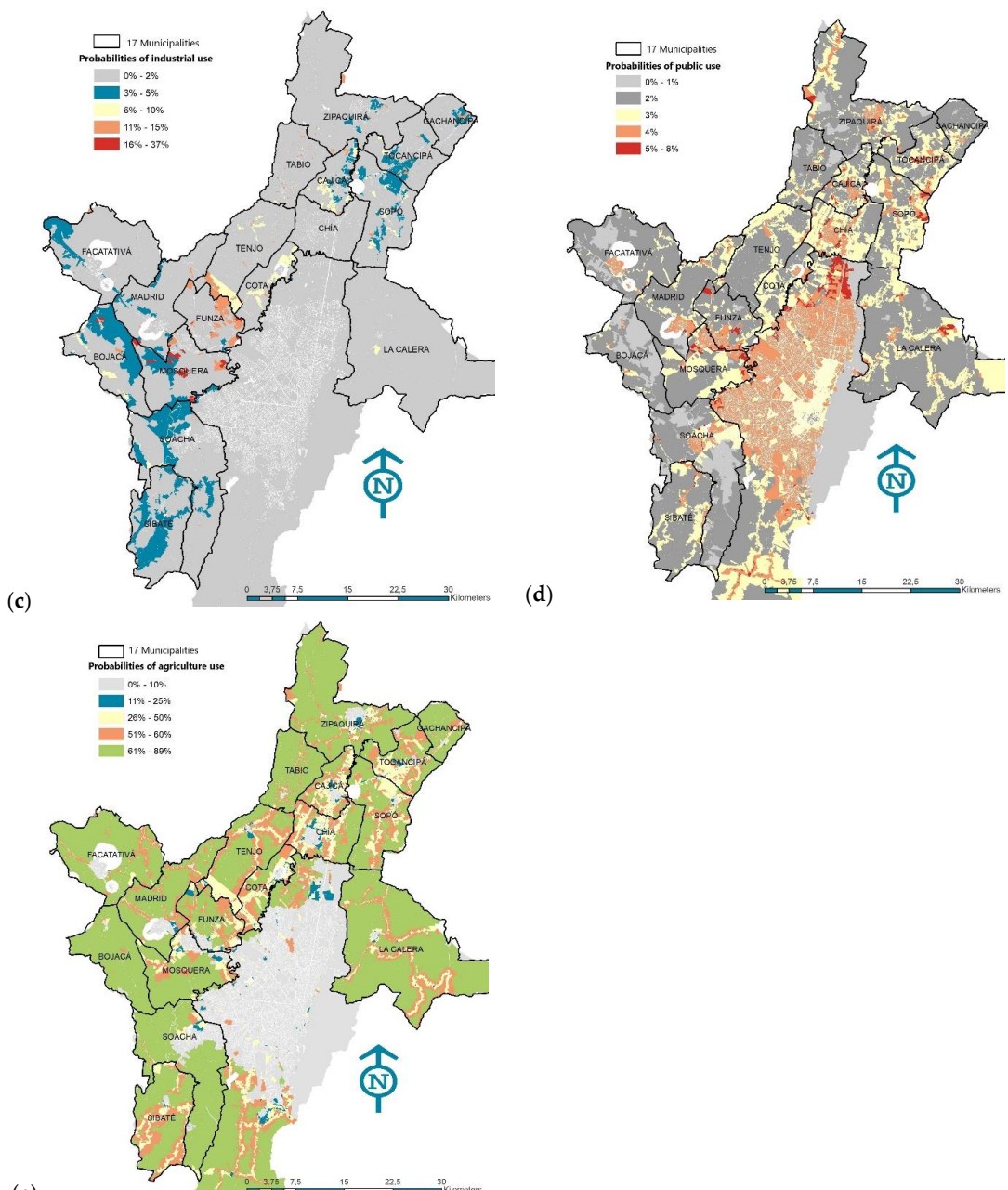

**Figure 5.** Probabilities estimated for each of the land use categories: (**a**) probabilities of residential use; (**b**) probabilities of commercial use; (**c**) probabilities of industrial use; (**d**) probabilities of public use; and (**e**) probabilities of agricultural use.

The general result can be summarized as follows: the accessibility pattern is not sufficient to identify the spatial structure in terms of municipal economic focus; and, by using the distribution of land uses, through its probability of occurrence controlled by accessibility, a more precise analysis can be performed of the layout of urban centers in the region and the possibilities of locating activities according to their position in the regional structure.

In this way, it is possible to observe how the region has different urban centers profiles; the main nucleus where commercial and public uses predominate; the commercial municipalities to the north; and the municipalities oriented to industry that are mainly to the west (although with differing importance and the possibility for this activity to become more attractive).

As mentioned above, land use models are related to classical theories of land rent. From this perspective, each land unit is assigned to the use that represents the highest rent, considering its specific attributes and location. For example, in the simplest version represented by a monocentric city model that determines the mobility pattern of households from place of residence to place of work, it is exactly the distance traveled (cost of transportation) that determines land rent prices.

This model explains that households optimize their location by evaluating accessibility to the CBD and the land rent. The resulting equilibrium is a land use pattern represented by concentric rings of residential use around the CBD, where the density decreases as the distance from the CBD increases. This is reflected in the signs associated with the coefficients estimated in the multinomial regression model.

The estimation of land use probabilities using the multinomial model enables municipal land use distribution to be identified from the regional spatial structure. In this sense, it is important for the analysis to determine the composition of urban centers in the study region, in terms of the distribution of activities reflected in the different categories of land use included.

### 4.2. Sensitivity Analysis: Expansion Areas and Suburban

In order to explore the expected changes in land use in zones with the greatest concentrated availability for future development, this section provides the results (sensitivity analysis) of two hypothetical normative changes on land use predicted by the models presented in the previous section.

The first exercise considers the re-categorization of suburban areas, actually outside the urban perimeter with normative restrictions concerning the density of constructions (it concerns 9630 spatial units, see suburban areas in Figure 1) towards the category of urban areas (where the normative uses and densities are higher).

Table 4 shows a comparison of the estimated probabilities of the model against a situation in which the spatial units mentioned are considered as urban areas. The upper part of Table 4 shows the probabilities associated with each use estimated under the current situation, whereas the lower panel shows the results simulated by the proposed change. The main changes observed are for the probability of residential use (which increases from 0.52 to 0.94) and for agricultural use (which decreases from 0.38 to 0). The probability of commercial and industrial use also decreases, whereas public use increases. These simulation results, and in particular, the resulting predominance of residential use, invite examination of regulatory alternatives that would incentivize a greater mix of uses in areas of future development. This simulation analysis concerns 9630 spatial units; therefore, it is possible calculate the standard deviation, the minimum and maximum values.

The second sensibility exercise focuses on the areas of urban expansion (3823 spatial units, included in the urban area in a particular category that permit their construction under specific circumstances such as the extension of public service infrastructure). The estimated probabilities of the model against a simulated situation in which the mentioned spatial units are converted into urban areas (Table 5), show the increase in the probability of residential use from 0.69 to 0.94, whereas the probability of agricultural use is reduced from 0.215 to 0.

**Table 4.** Change in probabilities of use following the re-categorization of suburban areas to urban.

| Variable | Mean Average | Std. Dev. | Min | Max |
|---|---|---|---|---|
| Residential Probability | 0.520 | 0.088 | 0.196 | 0.783 |
| Commercial Probability | 0.022 | 0.026 | 0.000 | 0.404 |
| Industrial Probability | 0.059 | 0.038 | 0.002 | 0.210 |
| Public Probability | 0.018 | 0.004 | 0.007 | 0.039 |
| Agricultural Probability | 0.381 | 0.094 | 0.041 | 0.665 |

**Table 4.** *Cont.*

| Variable | Mean Average | Std. Dev. | Min | Max |
|---|---|---|---|---|
| **Results of Simulation** | | | | |
| Residential Probability | 0.941 | 0.018 | 0.690 | 0.966 |
| Commercial Probability | 0.018 | 0.019 | 0.000 | 0.284 |
| Industrial Probability | 0.003 | 0.003 | 0.000 | 0.024 |
| Public Probability | 0.037 | 0.004 | 0.025 | 0.059 |
| Agricultural Probability | 0.000 | 0.000 | 0.000 | 0.001 |

**Table 5.** Change in probabilities of use following the re-categorization of areas of expansion to urban.

| Variable | Mean Average | Std. Dev. | Min | Max |
|---|---|---|---|---|
| Residential Probability | 0.687 | 0.055 | 0.359 | 0.789 |
| Commercial Probability | 0.056 | 0.064 | 0.000 | 0.603 |
| Industrial Probability | 0.005 | 0.003 | 0.000 | 0.016 |
| Public Probability | 0.038 | 0.005 | 0.020 | 0.058 |
| Agricultural Probability | 0.215 | 0.079 | 0.016 | 0.588 |
| **Results of Simulation** | | | | |
| Residential Probability | 0.939 | 0.022 | 0.681 | 0.962 |
| Commercial Probability | 0.019 | 0.024 | 0.000 | 0.289 |
| Industrial Probability | 0.005 | 0.003 | 0.000 | 0.027 |
| Public Probability | 0.037 | 0.003 | 0.024 | 0.055 |
| Agricultural Probability | 0.000 | 0.000 | 0.000 | 0.001 |

The results of these simulation exercises show how changes in regulations are incorporated in decisions by economic agents about location, through their willingness to pay for the use of the land. These facts are also reflected in the probabilities of land use.

## 5. Discussion

The literature has dealt with issues associated with the extension and growth of urban areas [22,26,57]. This document presents evidence on which could be the main variables that explains this growth. Additionally, this document makes it possible to evaluate how and why areas of economic activity are gaining place within urbanized areas. A very relevant aspect in this context is the potential implication regarding the urbanization of agricultural rural areas. In the specific case of this study region, historically, urban expansion and land occupation have developed in a disorderly manner.

The overall results can be summarized as follows: the accessibility pattern alone is not enough to identify the spatial structure in terms of the municipal economic focus; and, if the distribution of land uses is used (through use probability controlled by accessibility), there is a more precise analysis of the layout of urban centers in the region and the possibilities of locating activities according to their position in the regional structure.

Meanwhile, specific results can be summarized as follows: estimates of land use show that the probability of residential use is relatively high in urban areas in municipalities, as is to be expected. In the case of commercial use, Bogotá concentrates the greatest probabilities of use with concentrations in the predominant areas of economic activity. What is striking in this case is the extensive appearance of this use in the municipalities of Chía, Mosquera, Soacha and Zipaquirá. In the case of industrial use, a pattern of probability is presented which is dominated by the municipalities of Mosquera, Soacha, Funza and Cota. In addition, possible patterns of occurrence are presented in municipalities further

from the main nucleus such as Sibaté, Bojacá, Facatativá, Cajicá, Sopó, Tocancipá and Gachancipá. Finally, for public and agricultural uses, the following behavior is evidenced: a greater probability is observed in urban areas for the former, whereas for the latter, areas are observed outside the urban areas and far from main roads.

In such a way, the evidence presented in this document enabled us to identify these dynamics. Therefore, policy makers can incorporate these conditions when making transcendental decisions in the territories.

## 6. Conclusions

The evolution of monocentric regions towards polycentrism has been studied in developed countries. However, in the case of regions in developing countries, where there has been urbanization without economic growth, it is interesting to explore why monocentric structures may persist in regions or where the transition to polycentrism has been faster but less functional than in developed countries.

The regional analysis conducted in this document covered Bogotá and 17 municipalities. The study region was not jointly organized or planned; therefore, most of the phenomena observed in the region are largely due to dynamics generated by the economic influence of Bogotá and the local interests of attracting economic activities and residential land use in the regional structure as a whole.

This document aimed to analyze how the spatial structure of the region influences land occupation and land use distribution through levels of spatial accessibility, calculated as the sum of total employment weighted by distance. Utilizing this measure, it is possible to identify the pattern of land uses in the region and to also identify the differences in the intra-regional centers.

The regional accessibility pattern indicates the concentration of employment, and therefore, economic activity in the regional space. As expected, the highest indicator values were found in Bogotá as the main city, explaining most of the agglomerations that have previously been identified in the literature; specifically, business centers, industrial and commercial areas, and the airport.

The results of our estimations present evidence on the probability of residential use, which is high in the interior of urban areas in municipalities, as is to be expected. In the case of commercial use, Bogotá concentrates the greatest probabilities of use, with concentrations in the predominant areas of economic activity. In the case of industrial use, a pattern of probability is presented which is dominated by the municipalities of Mosquera, Soacha, Funza and Cota. Additionally, an interesting result concerns land of agricultural uses, where the transport infrastructure displaces the agricultural frontier, because it generates incentives to locate economic or residential activity.

This behavior, which reflects the agents' location decisions, is interpreted according to what is modeled in the theoretical framework of references that were presented in the corresponding sections of this document. However, solutions to land occupation problems do not necessarily work in all urban areas of the world. Indeed, the implications of the location of economic activity with respect to residential activity have objective implications: workers should be able to reach their workplaces in a relatively short time, e.g., 30 min; and the construction of transport infrastructure generates incentives for economic agents to occupy the land in strategic locations along said infrastructure.

Finally, it is important to mention that the behavioral dynamics of economic agents that are reflected in land occupation are a fundamental input for policy makers. In such a way, the empirical evidence presented in this document is aimed at enriching the debate on this matter.

**Author Contributions:** Conceptualization, A.D., H.E.S., and N.C.B.; methodology, A.D.; software, A.D.; validation, H.E.S., and N.C.B.; formal analysis, A.D.; investigation, H.E.S., and N.C.B.; data curation, A.D.; writing—original draft preparation, A.D.; writing—review and editing, H.E.S., and N.C.B. All authors have read and agreed to the published version of the manuscript.

**Funding:** The manuscript was funded by the Sergio Arboleda University and the Bogotá District Secretary for Planning through Agreement 369 of 2018: The development of a simulator to model the occu-pation of the city in the Bogotá D.C. region of the Sistema General de Regalías, BIPN code 2016000100031.

**Data Availability Statement:** From the Bogotá District Secretary for Planning (http://www.sdp.gov. co/content/visor-encuesta-de-esestantación-economicos, accessed on 12 August 2021), the information was taken from the Bogotá-Regional Economic Enterprise Survey (EEERB); from IDECA (https://www.ideca.gov.co/, accessed on 12 August 2021), the cadastral information of Bogotá was taken; and from the IGAC (https://www.igac.gov.co/, accessed on 12 August 2021), the cadastral information of the municipalities of Cundinamarca was obtained.

**Conflicts of Interest:** The authors declare no conflict of interest.

## Appendix A

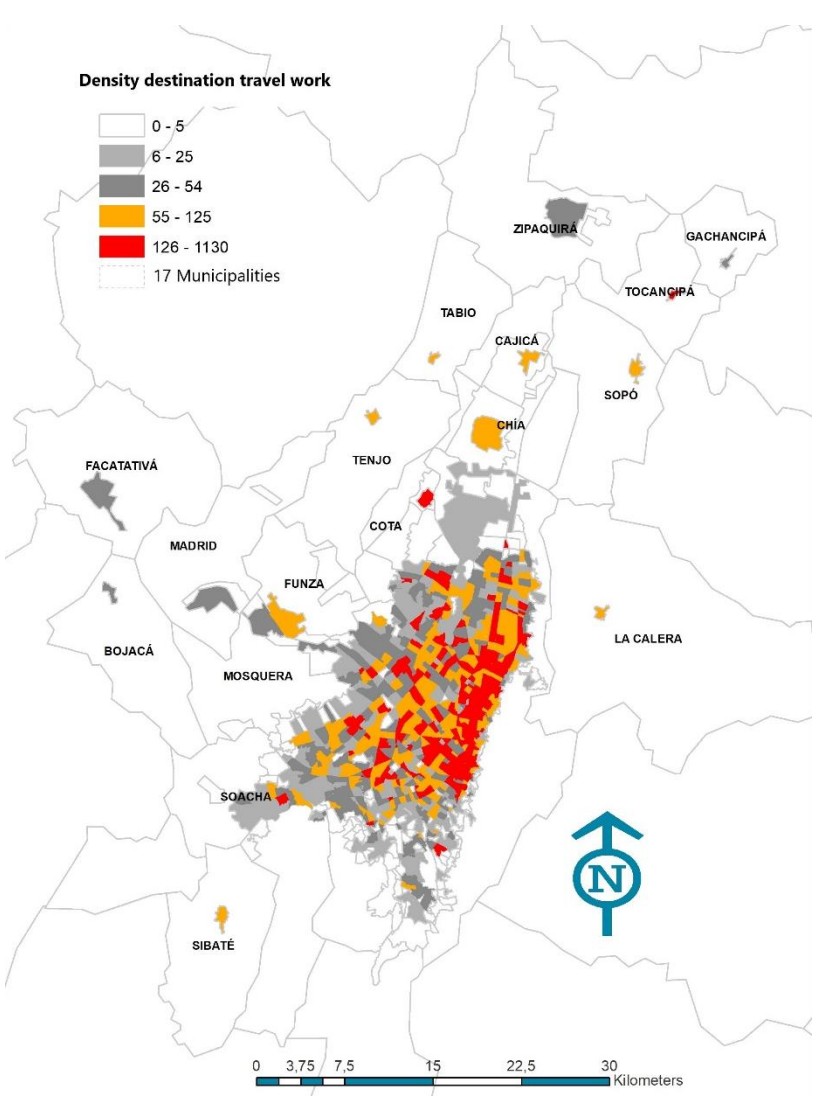

**Figure A1.** Density of trips to work per hectare. Source: Calculations made with information from Mobility Survey (2015).

```
Ho: Odds (Outcome-J vs Outcome - K) are independent of other alternatives
```

| Omitted | lnL(full) | lnL(omit) | chi2 | df | P>chi2 | evidence |
|---------|-----------|-----------|--------|----|--------|----------|
| 5 | -2.01e+04 | -2.01e+04 | 42.738 | 30 | 0.062 | for Ho |
| 6 | -2.56e+04 | -2.56e+04 | 37.062 | 30 | 0.175 | for Ho |
| 7 | -2.18e+04 | -2.18e+04 | 28.178 | 30 | 0.561 | for Ho |
| 8 | -1.39e+04 | -1.39e+04 | 41.717 | 30 | 0.076 | for Ho |

**Figure A2.** Small–Hsiao test to check independent alternatives.

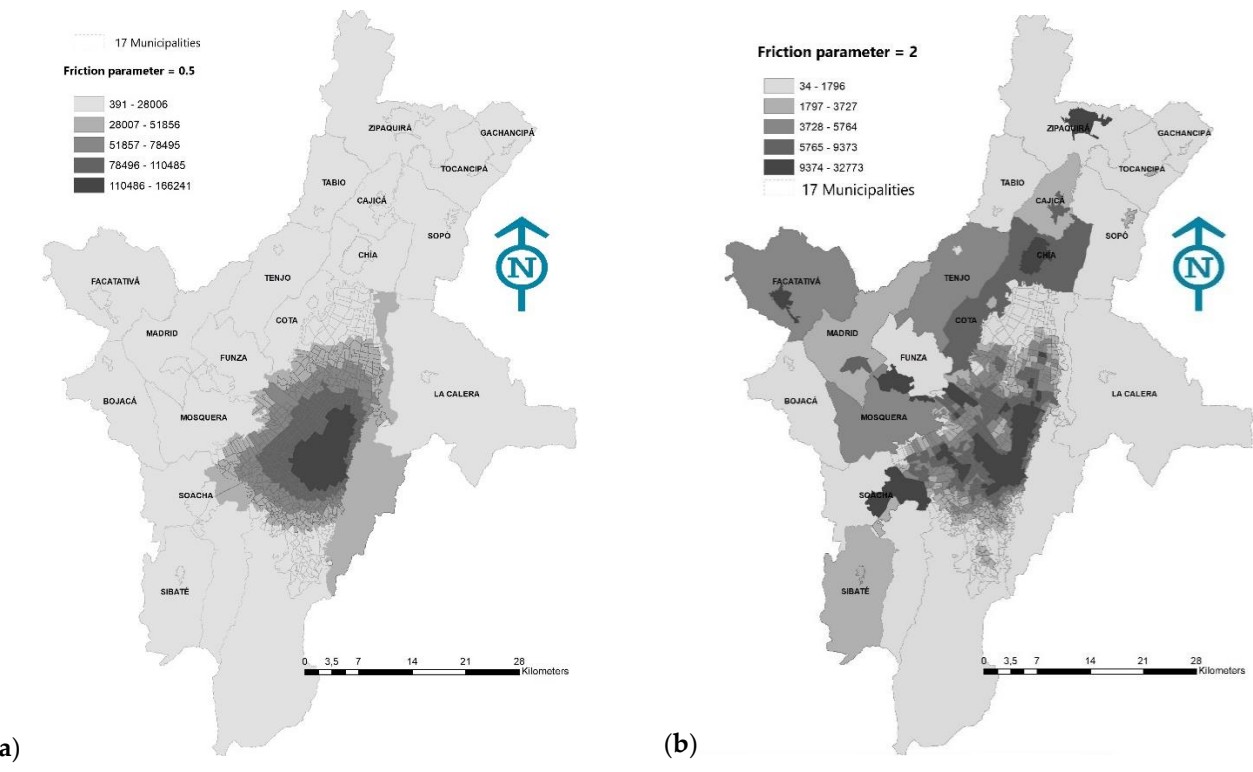

(**a**)  (**b**)

**Figure A3.** Changes in the accessibility indicator according to the friction parameter: (**a**) friction parameter equal to 0.5, (**b**) friction parameter equal to 2. Source: Based on information from the EEEBR (2017).

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
