# Peer review of "Regional Spatial Structure and Land Use: Evidence from Bogotá and 17 Municipalities"

_land, doi:10.3390/land10090908_

Round 1

Reviewer 1 Report

Most of my comments were taken into account and incorporated into the new work. It seems to me that the work can be published in its current form.

Minor corrections would certainly improve its quality, but do not cause a bad reception of the work (I refer the authors to follow all the comments that related to the previous version of the work). 

For example:

  • this is not necessary but could convert the table drawing in Apendix to text.
  • I still believe that "The description is too little detailed. The authors refer to the theory of land rent and the theory of rings, but do not explain how the results confirm their conclusions" (line 530).

Author Response

R/ The new version of the manuscript has an improvement on the second point, and the corrections suggested by the other referees on introduction, discussion of results and conclusions.

Reviewer 2 Report

The corrections made have significantly improved the quality of the manuscript. 
I make no comments now.

Author Response

The corrections made have significantly improved the quality of the manuscript. 
I make no comments now.

R/ Thanks!

Reviewer 3 Report

This paper tries to identify and solve in important issue. The methodology is sound. However, the paper provides too many pieces of information without effectively provide a reasonable link among them and explain how this information could be used in answering the research question and for better land use planning for the future of the study region. I have some major concerns listed below:

Title: Could it be “Regional spatial structure and land use: evidence from Bogotá and 2 17 municipalities, Colombia”?

Abstract: It should be improved with much clearer background, rationale of the study, methodology, results and conclusion.

Keywords: “Agglomeration economies” cannot be found from title and abstract.

Introduction: what is the research gap from your introduction? What is regional spatial structure? These terms have been repeatedly mentioned in the paper from the very beginning but have not been defined and explained in details toward the end of the paper. This is somehow confusing for the readers.

“In specific terms, this document seeks to explain the distribution of land uses in Bogotá and its 17 surrounding municipalities through various determinants, giving priority to the level of accessibility as an expression of the regional spatial structure.” Why did you choose Bogotá and its 17 surrounding municipalities? Why not introduce the very 17 surrounding municipalities to readers?

“1.1. Literature Review” a lonely sub-headline cannot be placed in a lonely section. Besides, it is very poor with limited literature about Spatial structure; Accessibility; Land uses; Land use planning etc. The current literature review is very confusing, and it is a collection of literature.

Materials and Methods: “These models, heavily influenced by Von Thunen's model (1827)”, how did these models heavily influenced by Von Thunen's model? What is Von Thunen's model? Equation 1-6 should be placed at method section.

“Characteristics specific to Bogotá and the region”. why did Bogotá and 17 municipalities change as Bogotá and the region? Is it 17 municipalities or the region?

“and 17 neighboring municipalities (with which Bogotá has been shown to have a commuting relationship at different levels)” what are the different levels?

Why did you only introduce population, urban area of Bogotá?

“2.2. Characteristics specific to Bogotá and the region” should be placed at results section.

Results: How come we get to variables in Table 2. Estimation results of the Multinomial Model of land use?

Please provide the parameters of the model error test.

Discussion: some interesting insights but still lack of coherency with the introduction/literature. still simply copy paste from commissioned works. Please add the contribution, deficiencies and prospects for future research of this article.

Conclusion: Conclusion mostly looks like a summary of the work done and the results obtained. No interpretation of the results in given as well as no recommendation for the government and policy makers as to how the results could be used.

Resolution of Figure 5 should be improved.

Some literature to consult:

Coupling Coordination Relationships between Urban-industrial Land Use Efficiency and Accessibility of Highway Networks: Evidence from Beijing-Tianjin-Hebei Urban Agglomeration, China. Sustainability. 2019; 11(5):1446. https://doi.org/10.3390/su11051446

A multi-dimensional multi-level approach to measuring the spatial structure of U.S. metropolitan areas. J. Transp. Land Use 2018, 11, 49–65.  DOI: https://doi.org/10.5198/jtlu.2018.893 

Author Response

We appreciate the valuable comments and contributions of the evaluators.

We hope that this new version will be improved, and we await your comments.

The responses to the corresponding evaluator are listed in the word file.

This manuscript is a resubmission of an earlier submission. The following is a list of the peer review reports and author responses from that submission.

Round 1

Reviewer 1 Report

The aim of the article was to analyze the influence of the spatial structure of the region on the occupation and distribution of land use through the levels of spatial accessibility, calculated as the sum of total employment weighted by distance (lines 619-621). It seems to me that the goal has not been achieved. The topic of the thesis is interesting, but the way of presenting the results is difficult to accept. The article is written in a way that makes its reception difficult. The work is too general, which makes it difficult to understand the message. The work requires ordering because the Authors mixed up the content of the chapters. Each chapter of the work is a partial repetition of the previous one. At least initially. The work should be organized so that the reader in one chapter gets the methodology, literature review, etc.

The article cannot be adopted in its current form.

Please find below detailed comments:

  1. The feature of occupation availability is not well-matched. The sprawl of the city is not only about occupation  availability. I guess it's more about residence outside the city center.
  2. Lines 99, 129: it seems to me that the problems that the researchers dealt with in these works should be described in detail. (I found it only in the summary).
  3. Lines 106-108: what are the exact measures?
  4. Lines 130-173: are an extension of the introduction and should probably be included in it.
  5. Line 225: what is q, n with a caret and q with a caret
  6. Line225: What are the first order conditions. What the authors wanted to define here.
  7. Formula 3: What does the symbol "l" mean
  8. Formula 4: What does the "K" symbol mean
  9. Are the transport costs in formula 5 not borne by the company? or not by the employee?
  10. Chapter 2 contains formulas that do not explain the model that the authors refer to. Can an explicit form of the model be written?
  11. Figure 1 is illegible. Geographic boundaries are indistinguishable from roads. There are no designated municipalities.
  12. Figure A1, in the Appendix, is called Figure 1
  13. I would transfer line 304- (reviewer's suggestion) to the literature review
  14. Lines 349-352 Is this the authors' contribution to the work?
  15. Line 376 no reference to the literature "known in the literature ..."
  16. Lines 407-424 are up to the results
  17. Lines 395-398 has no reference to literature
  18. As the authors understand the concept, spatial unit and location in formula 6. If it is the same, the indices in formula 6 have been poorly described: k and j
  19. Line 380: Is it a traditional weight matrix found in spatial analyzes?
  20. Formula 6 - how does this relate to defining a location as s?
  21. Marking (s) as profit in a given location may be confusing, I would suggest introducing an additional marking, e.g. D (s)
  22. Formula 7: In the data or methodology, the authors should accurately describe what variables the model contains
  23. The description of Table 2 is not detailed. It is not known why two models appeared. What do individual variables mean, etc.
  24. Lines 494-495 CO means that "there is a high probability of using ..."
  25. Formula 8: no description of symbols. Why is there a new pattern and what do the indices mean? How does formula 8 to 7 compare
  26. Why in Table 2 ('.' and ',') is used as the decimal separator? Conclusion: Why is the number of observations not an integer?
  27. Table 3: Are there job availability probabilities in this table? Have I lost my understanding of what is in the work?
  28. Lines 525-562 are understandable only for the Authors. The description is too little detailed. The authors refer to the theory of land rent and the rings theory, but do not explain how the results support their conclusions.
  29. Row 567-569 What are the 3 suburban types? Is it the same job?
  30. Table 4. What is the probability and how was it calculated? Why were the mean, standard deviation min and max compiled?
  31. Chapter 3.2 is redundant or needs better description.
  32. Line 571 where is map 1? 
  33. What is the Appendix?  The authors do not actually refer to it. 

Reviewer 2 Report

This article, seems to me, be correctly arranged and clearly shows the analysis of the land occupation and land use distribution in the wider area of Bogota. My suggestion is to implement two images and one table from the appendix in the article itself.

Reviewer 3 Report

Thank you for the opportunity to read some interesting research. However, the article needs additional clarification and references. 
1. (line 24) ".... the sum of the number of jobs weighted by distance" - from what distatns ? 
2. line 85 - why do you think that literature items 44-46 contribute most ? These are items from the 1960s. Today we have different methods of information transfer/communication and thus there are different priorities. 
In cities like Bogotá the management system and the importance of communication is changing 
3. line 210 - "...per unit of land" - what part of the land is meant? Are they neighbourhoods, street quarters ?  Are the surveyed parts equal to each other ?
4. Line 215-216 - again you use the term "... the distance to each of these locations". - How do you measure the distance ? (between what points). Where do you think the centre of the CBD is - the geometric centre of the area or the location of the selected building ? 
5. line 231-232 - how was the distance between the workplaces and the location of each employee determined ? 
6. Line 323 - are the Urban and Rual zones zones separated by administrative boundaries ? Are the Rual zones the area of 17 municipalities ? 

The method proposed by the authors requires access to multilayer databases (as they themselves emphasise they use EEEBR databases) 
I would like to consider methods which do not require such a large amount of data but give equally precise results (articles: Entropy of the Land Parcel Mosaic as a Measure of the Degree of Urbanization and Changes in Land Plot Morphology Resulting from the Construction of a Bypass: The Example of a Polish City). 
What advantages the proposed method has over the methods proposed by other researchers.